# Variation in hospital admission in febrile children evaluated at the Emergency Department (ED) in Europe: PERFORM, a multicentre prospective observational study

**Dorine M. Borensztajn**[1☉]*, **Nienke N. Hagedoorn**[1☉], **Irene Rivero Calle**[2‡], **Ian K. Maconochie**[3‡], **Ulrich von Both**[4‡], **Enitan D. Carrol**[5,6‡], **Juan Emmanuel Dewez**[7‡], **Marieke Emonts**[8,9,10‡], **Michiel van der Flier**[11,12,13‡], **Ronald de Groot**[14‡], **Jethro Herberg**[3‡], **Benno Kohlmaier**[15‡], **Emma Lim**[8‡], **Federico Martinon-Torres**[2‡], **Daan Nieboer**[16‡], **Ruud G. Nijman**[3‡], **Marko Pokorn**[17‡], **Franc Strle**[17‡], **Maria Tsolia**[18‡], **Clementien Vermont**[19‡], **Shunmay Yeung**[7‡], **Dace Zavadska**[20‡], **Werner Zenz**[15‡], **Michael Levin**[3‡], **Henriette A. Moll**[1‡], on behalf of PERFORM consortium: Personalised Risk assessment in febrile children to optimise Real-life Management across the European Union[¶]

**1** Department of General Paediatrics, Erasmus MC-Sophia Children's Hospital, Rotterdam, The Netherlands, **2** Genetics, Vaccines, Infections and Pediatrics Research Group (GENVIP), Hospital Clínico Universitario de Santiago de Compostela, Santiago de Compostela, Spain, **3** Section of Paediatric Infectious Diseases, Imperial College of Science, Technology and Medicine, London, United Kingdom, **4** Division of Paediatric Infectious Diseases, Dr. von Hauner Children's Hospital, University Hospital, Ludwig, Ludwig-Maximilians-Universität (LMU), München, Germany, **5** Institute of Infection and Global Health, University of Liverpool, Liverpool, United Kingdom, **6** Alder Hey Children's NHS Foundation Trust, Liverpool, United Kingdom, **7** Faculty of Tropical and Infectious Disease, London School of Hygiene and Tropical Medicine, London, United Kingdom, **8** Great North Children's Hospital, Paediatric Immunology, Infectious Diseases & Allergy, Newcastle upon Tyne Hospitals NHS Foundation Trust, Newcastle upon Tyne, United Kingdom, **9** Institute of Cellular Medicine, Newcastle University, Newcastle upon Tyne, United Kingdom, **10** NIHR Newcastle Biomedical Research Centre Based at Newcastle upon Tyne Hospitals NHS Trust and Newcastle University, Newcastle upon Tyne, United Kingdom, **11** Pediatric Infectious Diseases and Immunology, Wilhelmina Children's Hospital, University Medical Center Utrecht, Utrecht, The Netherlands, **12** Pediatric Infectious Diseases and Immunology, Amalia Children's Hospital, Radboud University Medical Center, Nijmegen, The Netherlands, **13** Department of Laboratory Medicine, Section Pediatric Infectious Diseases, Laboratory of Medical Immunology, Radboud Institute for Molecular Sciences, Radboud University Medical Center, Nijmegen, The Netherlands, **14** Stichting Katholieke Universiteit, Radboudumc Nijmegen, Nijmegen, The Netherlands, **15** Department of General Paediatrics, Medical University of Graz, Graz, Austria, **16** Department of Public Health, Erasmus University Medical Center Rotterdam, Rotterdam, The Netherlands, **17** Department of Infectious Diseases, University Medical Centre Ljubljana, Univerzitetni Klinični Center, Ljubljana, Slovenia, **18** Second Department of Paediatrics, National and Kapodistrian University of Athens, P. and A. Kyriakou Children's Hospital, Athens, Greece, **19** Department Pediatric Infectious Diseases & Immunology, Erasmus MC-Sophia Children's Hospital, Rotterdam, The Netherlands, **20** Department of Pediatrics, Rīgas Stradiņa Universitāte, Children Clinical University Hospital, Riga, Latvia

☉ These authors contributed equally to this work.
‡ These authors also contributed equally to this work.
¶ Membership of the PERFORM Consortium is provided in S1 Acknowledgments.
* d.borensztajn@erasmusmc.nl



**Data Availability Statement:** An anonymized data set necessary to replicate our study findings will be

## Abstract

### Objectives

Hospitalisation is frequently used as a marker of disease severity in observational Emergency Department (ED) studies. The comparison of ED admission rates is complex in

provided as a public repository with a DOI in case of acceptance of our manuscript.

**Funding:** This project was funded by the European Union's Horizon 2020 research and innovation programme to ML (Grant No. 668303), the NIHR Newcastle Biomedical Research Centre at Newcastle Hospitals NHS foundation trust to ME, and the National Institute for Health Research to RGN (CL-2018-21-007). The funders had no role in study design, data collection and analysis, decision to publish, or preparation of the manuscript.

**Competing interests:** The authors have declared that no competing interests exist.

**Abbreviations:** APLS, Advanced Paediatric Life Support; CRP, C-reactive protein; CSF, Cerebrospinal fluid; ECDC, European Centre for Disease Prevention and Control; ED, Emergency Department; FWF, Fever without focus; GI, gastro-intestinal infections; IQR, Interquartile range; LRTI, lower respiratory tract infections; MOFICHE, Management and Outcome of Fever in children in Europe; NICE, National Institute for Health and Care Excellence; PERFORM, Personalised Risk assessment in Febrile illness to Optimise Real-life Management across the European Union; PEWS, Paediatric Early Warning Score; PICU, Paediatric Intensive Care Unit; URTI, Upper respiratory tract infections.

potentially being influenced by the characteristics of the region, ED, physician and patient. We aimed to study variation in ED admission rates of febrile children, to assess whether variation could be explained by disease severity and to identify patient groups with large variation, in order to use this to reduce unnecessary health care utilization that is often due to practice variation.

## Design

MOFICHE (Management and Outcome of Fever in children in Europe, part of the PERFORM study, www.perform2020.org), is a prospective cohort study using routinely collected data on febrile children regarding patient characteristics (age, referral, vital signs and clinical alarming signs), diagnostic tests, therapy, diagnosis and hospital admission.

## Setting and participants

Data were collected on febrile children aged 0–18 years presenting to 12 European EDs (2017–2018).

## Main outcome measures

We compared admission rates between EDs by using standardised admission rates after adjusting for patient characteristics and initiated tests at the ED, where standardised rates >1 demonstrate higher admission rates than expected and rates <1 indicate lower rates than expected based on the ED patient population.

## Results

We included 38,120 children. Of those, 9.695 (25.4%) were admitted to a general ward (range EDs 5.1–54.5%). Adjusted standardised admission rates ranged between 0.6 and 1.5. The largest variation was seen in short admission rates (0.1–5.0), PICU admission rates (0.2–2.2), upper respiratory tract infections (0.4–1.7) and fever without focus (0.5–2.7). Variation was small in sepsis/meningitis (0.9–1.1).

## Conclusions

Large variation exists in admission rates of febrile children evaluated at European EDs, however, this variation is largely reduced after correcting for patient characteristics and therefore overall admission rates seem to adequately reflect disease severity or a potential for a severe disease course. However, for certain patient groups variation remains high even after adjusting for patient characteristics.

## Introduction

Febrile children form a large proportion of paediatric Emergency Department (ED) visits [1]; a substantial part of those children is admitted to the hospital for antibiotic treatment or observation due to diagnostic uncertainty [2, 3].

Besides having a significant adverse economic impact [3], admission can cause distress in children and caregivers [4] and efforts should be made to reduce unnecessary admissions.

Furthermore, a reduction of unnecessary admissions is essential in times of epidemics when resources are under strain.

A good starting point to reduce unnecessary hospital admission is by describing variation, as variation suggests potential overuse. A reduction in the variability of medical care is an important step in reducing health care costs [5] and studying practice variation in febrile children evaluated at the ED has been listed as a top research priority in a recent key publication [6].

Although several studies described variation in admission rates in children evaluated at the ED, most of these studies focused on specific patient groups such as children with bronchiolitis and only a minority corrected for disease severity [2, 7–10].

Furthermore, admission rates are commonly used as an outcome measure or proxy for disease severity [11–13] and evidence is needed to assess whether hospitalisation can be used as a valid proxy for disease severity, especially when admission is short or therapy is not escalated.

We aimed to study variation in hospital admission for febrile children at European EDs, in order to assess whether this variation can be explained by patient characteristics and disease severity, to identify patient groups with remaining large variation after correcting for patient factors, in order to be able to use this data for local hospitals to get insight in potential unnecessary admissions in specific patient groups.

## Methods

### Study design

This study is part of the MOFICHE study (Management and Outcome of Febrile children in Europe), which is embedded in the PERFORM study (Personalised Risk assessment in Febrile illness to Optimise Real-life Management across the European Union, www.perform2020.org). The MOFICHE study is a multicentre study evaluating the management and outcome of febrile children using routinely collected data.

The study was approved by the ethical committees of all the participating hospitals and no informed consent was needed for this study. Austria (Ethikkommission Medizinische Universitat Graz, ID: 28–518 ex 15/16), Germany (Ethikkommission Bei Der LMU München, ID: 699–16), Greece (Ethics committee, ID: 9683/18.07.2016), Latvia (Centrala medicinas etikas komiteja, ID: 14.07.201 6. No. Il 16–07–14), Slovenia (Republic of Slovenia National Medical Ethics Committee, ID: ID: 0120-483/2016-3), Spain (Comité Autonómico de Ética de la Investigación de Galicia, ID: 2016/331), The Netherlands (Commissie Mensgebonden onderzoek, ID: NL58103.091.16), United Kingdom (Ethics Committee, ID: 16/LO/1684, IRAS application no. 209035, Confidentiality advisory group reference: 16/CAG/0136).

In all the participating UK settings, an additional opt-out mechanism was in place. Patients were not directly involved in the design of this study.

### Study population and setting

Twelve EDs from eight different European countries (Austria, Germany, Greece, Latvia, the Netherlands, Spain, Slovenia, and the United Kingdom) participated. Participating hospitals are either university (n = 9) or large teaching hospitals (n = 3) and 10 EDs have paediatric intensive care facilities. Nine EDs are paediatric focused and three serve children and adults. Care for febrile children is supervised by general paediatricians (6 EDs), paediatric emergency physicians (2 EDs) or by either one (3 EDs). In ten hospitals a paediatric infectious diseases specialist is available. Data were collected between January 2017 and April 2018. All children aged 0–18 years presenting with fever (temperature > = 38.0˚C) or a history of fever in the previous 72 hours were included. Data collection ranged from one week per month to the entire month (S1 Table).

## Data collection

Data were entered into the patient's record as part of routine care by the treating physician and were then manually extracted from these records and manually entered into an electronic case report form (eCRf) by the research team. Data included hospital-level factors, general patient characteristics, markers of disease severity, diagnostic tests, therapy at the ED, diagnosis (working diagnosis/focus of infection, final diagnosis) [14] and disposition.

## Definitions

Hospital-level factors included variables that varied between hospitals and were expected to be related to admission: number of annual ED visits, level of supervision, availability of primary care during out of office hours regulations regarding the maximum time a patient could spend at the ED before having to be admitted or discharged [15, 16].

General patient characteristics included age, gender, comorbidity, time of presentation (e.g. daytime, weekend, night-time), referral, duration of fever, and medical care for the same complaint in the previous five days. Previous medical care included visits at the same or a different facility and was documented by the treating physician as part of the medical history and manually extracted from the patient's medical records by the research team.

Children were categorised into the following age categories: < 3 months, 3–12 months, 1–5 years, 5–12 years, >12 years. Comorbidity was defined as a chronic underlying condition that is expected to last at least 1 year.

Markers of disease severity included triage urgency category, vital signs, Paediatric Early Warning Score (PEWS) and presence of "red traffic light" symptoms for identifying risk of serious illness (alarming signs) (National Institute for Health and Care Excellence (NICE) guideline on fever [17]. Vital signs were classified as abnormal according to Advanced Paediatric Life Support (APLS) reference ranges [18].

We used the PEWS developed by Parshuram et al. [19] to assess the overall clinical status. PEWS included heart rate, capillary refill, respiratory rate, work of breathing, oxygen saturation and oxygen therapy. Blood pressure was excluded from the PEWS as it was not routinely performed in our study.

Diagnostic tests included any diagnostic test, C- reactive protein (CRP) (<20, 20–60,>60 mg/L) [14], any imaging, chest x-ray and blood and cerebrospinal fluid (CSF) cultures. Chest x-ray was defined as abnormal in the following cases: focal infiltrate or consolidation, diffuse abnormalities, pleural effusion or other abnormality).

Diagnostic tests results were included if the results were available during the ED visit and were either available as absolute number (e.g. CRP) or abnormalities were predefined (chest x-rays). Other tests were included in the analysis as performed/not performed (e.g. blood cultures) as these test results were not available at the ED and thus did not influence admission.

Treatment included oxygen, intravenous antibiotics and immediate lifesaving interventions (airway/breathing support, emergency procedures, haemodynamic support, emergency medications).

Final diagnosis was standardised according to a consensus-based flowchart [14, 20], that combines clinical data, CRP and cultures performed at the ED or within 24 hours after ED presentation. Patients were classified into presumed bacterial (definite or probable bacterial, bacterial syndrome), unknown bacterial/viral, presumed viral (definite or probable viral, viral syndrome) and other (S1 Fig). Focus of infection was classified into sepsis/meningitis, upper respiratory tract infection (URTI), lower respiratory tract infections (LRTI), urinary tract infections (UTI), gastro-intestinal infections (GI), skin/musculoskeletal infections, childhood exanthemas/flu-like illness and fever without focus (FWF).

Disposition was categorised into discharge, general ward admission < 24 hours, general ward admission ≥ 24 hours or Paediatric Intensive Care Unit (PICU) admission. Any admission was defined as general ward or PICU admission.

## Data quality

Data quality was improved and standardised by using a digital training module for treating physicians at the ED who assess febrile patients, in order to reduce missing values and improve uniform data quality, including the clarification of NICE alarming signs. Clinical data were entered into a standardised eCRF by trained research team members. Monthly teleconferences and biannual meetings were organised and quarterly data quality reports were discussed with all partners.

Patients with missing disposition and missing information on intravenous antibiotics were excluded from the analysis. Missing determinants such as heart rate and respiratory rate were handled by using multiple imputation.

## Data analysis

We used multilevel logistic regression with a random intercept for each ED to study variation of admission between hospitals. Determinants of admission used in this multilevel logistic regression included hospital-level factors (referral, supervision, primary care availability, regulations on time spent at the ED), general patient characteristics, markers of disease severity, diagnostic tests, therapy and working diagnosis (described in detail in Tables 1–3). We included variables if they improved the model defined by a likelihood ratio test using p<0.05 as a cut-off value.

We calculated standardised admission rates using indirect standardisation, where the expected number of admissions was standardised to the average ED. Standardised admission ratios are the ratio between observed and expected admissions in an ED. The expected number of admissions was estimated through the adjusted model, by summing the predicted probabilities from the adjusted model of admissions for each patient. Standardised rates >1 demonstrate higher rates than expected and rates <1 indicate lower rates than expected (S2 Fig). We first analysed overall variation between hospitals using the multilevel model described above.

Secondly, we analysed variation between hospitals for specific patient groups, such as age groups, final diagnosis and focus of infection. Standardised rates were visualised in a heat map [21].

Additionally, we fitted random effects logistic regression models to investigate if rates differed for each subgroup. We compared a random effects logistic regression model, including all variables used to estimate the admission rate as a fixed effect and a random intercept for hospital with different standard deviations for the different subgroups, with a random effects logistic regression model including the same covariates and one overall random intercept. Model fits were compared using a likelihood ratio test, and a significant effect indicates that variation in admission rates is different for each of the subgroups.

SPSS version 25 and R version 3.5.1 were used for data analysis. Imputation was performed by using the MICE package in R.

## Results

Of the total population of 38,480 patients, 360 patients were excluded due to missing disposition or missing information on intravenous antibiotics, leaving 38,120 children for analysis. Patient characteristics are described in Table 1.

The most common areas of infection were URTI (n = 19,947, 52.3%), LRTI (n = 5,621, 14.7%), GI (n = 3,958, 10.4%) and FWF (n = 2,950, 7.7%). The majority of children had a

**Table 1. Patient characteristics*.**

| N = 38.120 | n (%) | Range EDs % | Missing data n % |
|---|---|---|---|
| **Age in years,** median (IQR) | 2.8 (1.3–5.6) | | **0 (0.0)** |
| **Gender** | | | |
| **Male** | 20,910 (54.9) | 51.7–59.5 | 1 (0.0) |
| **Comorbidity** | 6.416 (17.0) | 5.2–65.6 | 364 (1.0) |
| **Complex** | 1.583 (4.2) | 0.4–32.4 | 364 (1.0) |
| **Medical care in the last 5 days** | 9,817 (26.7) | 11.1–40.6 | 1307 (3.4) |
| **Time of presentation** | | | 0 (0.0) |
| Daytime (8–17) | 19,400 (50.9) | 42.3–63.2 | |
| Evening (17–23) | 11,792 (30.9) | 24.7–36.6% | |
| Night-time (23–8) | 6,928 (18.2) | 9.2–29.6% | |
| Weekend | 12,453 (32.7) | 24.3–41.3 | |
| **Season** | | | 0 (0.0) |
| Winter | 13,521 (35.5) | 27.1–53.4 | |
| Spring | 9,618 (25.2) | 18.0–31.1 | |
| Summer | 6,180 (16.2) | 9.4–23.1 | |
| Autumn | 8,801 (23.1) | 7.2–32.5 | |
| **Referral** | | | 1159 (3.0) |
| Self | 20,976 (56.8) | 0.6–94.9 | |
| GP/community-based paediatrician | 6,350 (17.2) | 2.8–91.2 | |
| Emergency medical service (EMS)* | 5,563 (15.1) | 0.1–41.6 | |
| Other | 4,072 (11.0) | 0.4–45.5 | |
| **Triage urgency** | | | 1174 (3.1) |
| Low: non-urgent standard, | 23,774 (64.3) | 10.1–91.2 | |
| High: immediate, very urgent, urgent | 13,172 (35.7) | 8.8–89.9 | |
| **Vital signs (respiratory rate, heart rate, oxygen saturation)** | | | |
| Tachycardia | 9,529 (25.0) | 4.8–41.0 | 3461 (9,1) |
| Tachypnoea | 5,656 (14.8) | 0.7–33.4 | 8718 (22.9) |
| Low oxygen saturation ≤ 94% | 8.48 (2.2) | 0.6–5.3 | 5463 (14.3) |
| **Nice "red traffic lights" (alarming signs)** | | | |
| Decreased consciousness | 199 (0.5) | 0.1–3.6 | 375 (1.0) |
| Ill appearance | 5,985 (16.4) | 0.9–50.5 | 1693 (4.4) |
| Increased work of breathing | 757 (2.3) | 0.1–16.9 | 5465 (14.3) |
| Dehydration | 1,893 (6.1) | 0.4–15.9 | 6937 (18.2) |
| Age < 3 months | 1,049 (2.8) | 1.1–13.3 | 1049 (2.8) |
| Rash petechiae / non-blanching | 1,100 (3.3) | 1.5–7.7 | 4377 (11.5) |
| Meningeal signs | 135 (0.4) | 0.1–2.2 | 2015 (5.3) |
| Status epilepticus | 66 (0.2) | 0.0–2.0 | 1134 (3.0) |
| Focal neurology | 132 (0.4) | 0.0–3.2 | 2427 (6.4) |

*Patients referred by a health care provider that were brought in by EMS, were categorised as referral by EMS.

presumed viral infection (n = 21,448, 56.3%); presumed bacterial infections occurred in 8,516 children (22,3%), 5,848 children (15,3%) were classified as unknown and 2,308 (6,0%) children were classified as other.

9,695 children were admitted to a general ward (25.4%, range 5.1–54.5%) and 156 children (0.4%, range 0.1–4.0%) were admitted to the PICU, Table 2.

Unadjusted admission rates varied across EDs, ranging from 0.2–2.1. After adjustment, variability was reduced, with standardised admission rates ranging from 0.6–1.5 (Table 4, S2

**Table 2. Diagnostic tests, therapy at the ED and disposition\*.**

| N = 38.120 | n (%) | Range EDs % | Missing data n % |
|---|---|---|---|
| **Diagnostic tests** | | | |
| Any diagnostic test | 27,252 (71.5) | 42.7–100.0 | 0 (0.0) |
| Any blood test | 17,452 (45.8) | 9.6–92.7 | 0 (0.0) |
| CRP | 17,130 (45.1) | 7.7–92.2 | 0 (0.0) |
| Blood cultures | 3,531 (13.9) | 0.6–46.7 | 0 (0.0) |
| CSF tests | 448 (3.1) | 0.3–11.3 | 0 (0.0) |
| Any imaging | 6,908 (18.1) | 8.4–27.1 | 0 (0.0) |
| **Therapy** | | | |
| Immediate life-saving interventions | 640 (1.7) | 0.1–9.0 | 23 (0.1) |
| Oxygen therapy | 1,082 (2.8) | 0.6–13.9% | 150 (0.4) |
| Intravenous antibiotics | 3,777 (9.9) | 2.9–21.8% | 360 (0.9) |
| **Disposition** | | | 56 (0.1) |
| Discharged home | 28,051 (73.6) | 45.1–94.8 | |
| Left without being seen | 218 (0.6) | 0.0–2.0 | |
| General ward admission | 9,695 (25.4) | 5.1–54.5 | |
| Admission < 24 hours | 2,001 (20.6) | 0.0–16.3 | |
| Admission ≥ 24 hours | 7,229 (74.6) | 2.5–42.3 | |
| Admission duration unknown | 465 (4.8) | 0.0–6.5 | |
| Admission to ICU | 156 (0.4) | 0.1–4.0 | |

Table). Hospital-level factors were not significantly associated with admission and were excluded from the final model.

## Variation by type of admission

Differences between hospitals in short admission rates (0.1–4.2 unadjusted, 0.1–5.0 adjusted) and PICU admission rates (0.2–10.2 unadjusted, 0.2–2.2 adjusted) were most pronounced. Short admission rates were highest in the UK hospitals, while these centres had average rates for admissions ≥24 hours (Table 4).

## Subgroup variation

The least variation was seen in children with sepsis/meningitis (0.8–1.0 unadjusted, 0.9–1.1 adjusted), UTI (0.4–2.0 unadjusted, 0.7–1.5 adjusted), LRTI (0.4–1.5 unadjusted, 0.6–1.3 adjusted) and GI (0.3–2.1 unadjusted, 0.6–1.6 adjusted).

Larger variation was seen in children with URTI (0.1–2.9 unadjusted, 0.4–1.7 adjusted) and skin/musculoskeletal infections (0,3–1,7 unadjusted, 0,7–2,3 adjusted). The largest variation was seen in FWF (0.1–1.9 unadjusted, 0.5–2.7 adjusted, Table 5, S2 and S3 Tables).

**Table 3. Model correction factors.**

| Hospital-level factors | Referral, supervision, primary care, regulations on time spent at the ED\* |
|---|---|
| **Patient characteristics** | Age, gender, time of arrival, duration of fever, previous medical care, comorbidity |
| **Markers of disease severity** | Triage urgency, NICE red traffic lights, vital signs |
| **Diagnostic tests** | Any diagnostic tests, any blood tests, CRP, blood culture, imaging, CSF tests |
| **Therapy** | Life-saving interventions, oxygen, iv antibiotics |
| **Working diagnosis** | Focus of infection |

\* None of the hospital-level factors were significant and were therefore not included in the final model.

**Table 4. Heat map of standardised admission rates per hospital: All children\*.**

| Hospital | any admission | admission < 24 h | admission ≥ 24 h | PICU admission |
|---|---|---|---|---|
| Austria | 1.1 | 0.1 | 1.6 | 2.2 |
| Germany | 0.7 | 0.1 | 0.8 | 1.1 |
| Greece | 1.5 | 0.3 | 1.2 | 1.7 |
| Latvia | 1.0 | n.a. | 1.5 | 2.1 |
| NL, 1 | 0.9 | 1.2 | 0.9 | 1.8 |
| NL, 2 | 0.8 | 0.7 | 0.8 | 0.8 |
| NL, 3 | 1.1 | 1.7 | 1.3 | 1.8 |
| Slovenia | 1.2 | 1.1 | 1.4 | 1.6 |
| Spain | 0.6 | 1.4 | 0.5 | 1.4 |
| UK, Liv | 1.3 | 5 | 1.1 | 0.2 |
| UK, New | 1.2 | 4.7 | 0.9 | 0.6 |
| UK, Lon | 1.0 | 3.1 | 0.8 | 0.4 |

\* Corrected for patient characteristics, markers of disease severity, diagnostic tests, therapy and working diagnosis.

\*\* No admissions < 24 hours in Latvia.

Variation was larger in the unknown group (0.2–1.8 unadjusted, 0.5–2.4 adjusted) than in the presumed bacterial group (0.2–1.7 unadjusted, 0.6–1.3 adjusted) or presumed viral group (0.2–2.3 unadjusted, 0.6–1.3 adjusted, Table 6, S2 Table). A significant difference was found between these subgroups (p<0.001).

Variation was highest in children>5 years old (0.2–2. 1 unadjusted, 0.6–1.7 adjusted) and >12 years (0.2–1.9 unadjusted, 0.6–1.7 adjusted) and least in children <3 months (0.5–1.2 unadjusted, 0.7–1.3 adjusted, Table 7, S2 Table).

## Discussion

We found large differences in unadjusted admission rates between the participating EDs. Overall, variation largely diminished after correcting for general patient characteristics,

**Table 5. Heat map of standardised any admission rates per hospital for different patient groups: Focus of infection\*.**

| Hospital | sepsis/meningitis | URTI | LRTI | Fever without focus |
|---|---|---|---|---|
| Austria | 1.0 | 1.0 | 1.2 | 1.0 |
| Germany | 1.0 | 0.7 | 0.6 | 0.8 |
| Greece | 1.0 | 1.5 | 1.1 | 2.7 |
| Latvia | 1.1 | 1.1 | 1.1 | 1.1 |
| NL, 1 | 1.0 | 0.8 | 0.9 | 0.9 |
| NL, 2 | 1.0 | 0.6 | 0.8 | 0.7 |
| NL, 3 | 1.0 | 1.0 | 1.2 | 1.0 |
| Slovenia | 0.9 | 1.4 | 1.1 | 1.1 |
| Spain | 1.0 | 0.4 | 0.8 | 0.5 |
| UK, Liv | 1.0 | 1.7 | 1.3 | 1.1 |
| UK, New | 1.0 | 1.5 | 1.1 | 1.2 |
| UK, Lon | 0.9 | 0.8 | 1.2 | 1.0 |

\* Corrected for patient characteristics, markers of disease severity, diagnostic tests, therapy and working diagnosis.

\*\*URTI = upper respiratory tract infection.

\*\*\* LRTI = lower respiratory tract infection.

**Table 6. Heat map of standardised any admission rates per hospital for different patient groups: Final diagnosis*.**

| Hospital | Presumed bacterial | Unknown viral/bacterial | Presumed viral |
|---|---|---|---|
| Austria | 1.3 | 1.2 | 1.0 |
| Germany | 0.8 | 0.8 | 0.6 |
| Greece | 1.3 | 2.4 | 1.2 |
| Latvia | 0.9 | 1.2 | 1.1 |
| NL, 1 | 1.0 | 0.8 | 0.8 |
| NL, 2 | 0.8 | 0.8 | 0.7 |
| NL, 3 | 1.1 | 1.0 | 1.1 |
| Slovenia | 1.0 | 1.0 | 1.3 |
| Spain | 0.6 | 0.5 | 0.6 |
| UK, Liv | 1.2 | 1.5 | 1.3 |
| UK, New | 1.2 | 1.2 | 1.3 |
| UK, Lon | 0.8 | 1.0 | 1.0 |

* Corrected for patient characteristics, markers of disease severity, diagnostic tests, therapy and working diagnosis.

markers of disease severity, diagnostic tests, therapy and focus of infection, showing that, at least in part, the variation in admission rates is related to patient characteristics.

However, variation remained high in specific groups, such as URTI, FWF and the unknown viral/bacterial group.

The larger variation observed in specific patient groups might be related to a higher degree of diagnostic uncertainty. Furthermore, it is possible that there is more uniformity in guideline use in some clinical problems than in others. Our previous survey showed that most settings used a local or national guideline for febrile children and a minority used the NICE guideline, while in sepsis, around half of the settings used the NICE guideline [15].

Possible other explanations for these persistent differences include different physician practice patterns due to physician educational background or experience level [3, 8, 16, 22].

Furthermore, variation was higher for short admissions in comparison to longer admissions and variation in PICU admissions was higher than for any admission. Possible explanations include different admission criteria or different local ED regulations [15, 23, 24].

**Table 7. Heat map of standardised any admission rates per hospital for different age groups*.**

| Hospital | < 1 month | < 3 months | 3–12 months | 1–5 years | 5–12 years | >12 years |
|---|---|---|---|---|---|---|
| Austria | 0.5 | 1.2 | 1.2 | 1.0 | 1.0 | 1.2 |
| Germany | 1.3 | 0.9 | 0.7 | 0.6 | 0.6 | 1.0 |
| Greece | 1.3 | 1.3 | 1.4 | 1.5 | 1.7 | 1.7 |
| Latvia | 1.0 | 1.0 | 1.1 | 1.0 | 1.0 | 1.0 |
| NL, 1 | 1.0 | 0.9 | 0.9 | 0.9 | 0.9 | 0.9 |
| NL, 2 | 1.0 | 0.8 | 0.7 | 0.8 | 0.8 | 0.7 |
| NL, 3 | 1.0 | 1.0 | 1.1 | 1.2 | 1.1 | 1.0 |
| Slovenia | 1.2 | 1.2 | 1.2 | 1.2 | 1.1 | 1.1 |
| Spain | ** | 0.7 | 0.5 | 0.6 | 0.6 | 0.6 |
| UK, 1 | 1.0 | 1.2 | 1.4 | 1.4 | 1.2 | 1.1 |
| UK, 2 | 1.0 | 0.9 | 1.2 | 1.3 | 1.4 | 1.1 |
| UK, 3 | 0.9 | 1.0 | 0.8 | 1.0 | 1.0 | 1.1 |

* Corrected for patient characteristics, markers of disease severity, diagnostic tests, therapy and working diagnosis.

** insufficient data to be included.

For example, the three centres with the highest short-admission rates were all from the UK. These higher short-admission rates might be related to the four-hour target, which states that 95% of all ED patients should be discharged or admitted within 4 hours [23, 24]. Although several other settings had regulations on how long patients could stay at the ED before they had to be admitted or discharged, these were much longer than 4 hours and thus are less likely to influence (short) admission rates.

Additionally, our previous survey among the participating settings of the current study, showed different PICU admission criteria (e.g. the use of high flow oxygen) [15] and a UK study showed a wide variation in PICU admission rates and–similar to our data—highlight the need for clear PICU admission criteria [25].

Our data show that overall, differences between settings are largely reduced after correcting for differences in patient population and as such, overall admission rates reflect disease severity or the potential risk for a severe disease course, such as young febrile infants.

However, the fact that we found that for specific subgroups (e.g. upper respiratory tract infections, fever without focus) variation remained high even after correcting for patient characteristics suggests there is a large number of potential unnecessary admissions and there is room for improvement of the management of these patients. This is relevant year-round when in it comes to reducing avoidable health care costs, and even more relevant in peak seasons for hospitalisations, for example due to respiratory syncytial virus.

## Strengths and weaknesses

The strengths of this European multicentre study are the standardised data collection during all seasons in a large scope of febrile children. Secondly, we corrected for a large number of determinants, improving comparison between EDs.

The EDs participating in this study are university or large teaching centres and all but one have an onsite PICU and have paediatric infectious disease specialists available [20, 26] and therefore represent a selected standard of care, which might limit generalisability. However, the inclusion of over ten hospitals with differences in patient case mix as well as local and regional health care organisation, improves generalisability.

Moreover, additional factors that were not included in the analysis, such as socio-economic status [7, 8, 27–31], or cultural differences could possibly influence some of the variability. However, a study in children with asthma did not find an association between socio-economic status and PICU admission rates [30].

Furthermore, we did correct for a large number of important clinical confounders.

Although, the use of PEWS is mainly validated to be used as a change in scores in hospital settings and not a single value. However, several studies have shown a single PEWS at arrival at the ED to be associated with general admission, PICU admission and serious illness [31–33], and as such it was included in our study as a marker of disease severity.

Lastly, our study did not look at ED revisits and subsequent admission, but previous studies showed low revisit rates in febrile children [34].

## Findings in relation to literature

Although several studies have assessed variation in admissions rates, many did not correct for patient characteristics or disease severity [7, 8, 10, 35], while others only corrected for severity on a hospital-level [2], used general adjustment measures based on International Classification of Diseases codes [2] or corrected for a limited number of variables only [9, 36].

To our knowledge this is the first study comparing admission rates for febrile children in the entire paediatric age group while adjusting for a broad number of patient characteristics.

### Implications for clinical practice and research

The wide variation of hospital admissions in specific patient groups found in our study highlights differences in care and poses question about the compliance with (inter)national best care recommendations. This variation might reflect under- as well as overtreatment which can lead to unnecessary health care cost.

Our results could be used as a starting point for a more standardised approach by identifying patients with a large variation in admission rates.

The high variation in admission rates for children in certain patient groups, such as FWF, might be due in part to diagnostic uncertainty [3] which has been suggested to increase admissions. Additional methods to decrease diagnostic uncertainty, such as novel diagnostic tests that offer improved differentiation between self-limiting viral disease and bacterial infections, might reduce unnecessary admissions.

Furthermore, variation can be reduced by measures that improve guideline adherence [37, 38].

As our results showed higher short admission rates in settings that adhere to a 4-hour target, improving patient flow by accelerating test results could potentially reduce avoidable admissions, leading to a more cost-effective health care approach. There is evidence that admissions in general and short admissions in particular tend to increase during times of crowding, possibly admitting patients when there is diagnostic uncertainty [39]. Possible strategies to improve patient flow include the use of point of care tests [40], the introduction of general practitioners at the ED that can redirect low urgent patients from the ED [41], the implementation of observation units where admission can take place for a few hours only [42, 43] or by developing prediction models which can be used to identify children that need admission at an earlier stage [29].

Lastly, as previous studies have shown that paediatric emergency physicians are less likely to admit children than general emergency physicians and that the presence of a senior consultant is associated with a reduced rate of total as well as short-stay admissions, training physicians caring for febrile children can play an important part in reducing unnecessary admissions [22, 44].

## Conclusion

Large variation exists in admission rates of febrile children evaluated at European EDs. This variation is largely reduced after correcting for patient characteristics and therefore in general admission rates seem to adequately reflect disease severity or the potential risk for a severe disease course. However, for certain patient groups, such as children with fever without focus, variation remains high even after adjustment.

Focusing on patient groups with large variation in admission rates can be used as a starting point for a more uniform and cost-effective health care approach.

## Supporting information

**S1 Fig. Categorisation of presumed cause of infection.**
(PDF)

**S2 Fig. Graphic representing standardised admission rates.**
(PDF)

**S1 Table. Hospital characteristics.**
(PDF)

**S2 Table. Heat map data and 95% confidence interval.**
(PDF)

**S3 Table. Heat map of standardised any admission rates per hospital for different patient groups: Focus of infection, remaining diagnoses**\*. \* Corrected for patient characteristics, markers of disease severity, diagnostic tests, therapy and working diagnosis. \*\* UTI = urinary tract infection.
(PDF)

**S1 Acknowledgments. PERFORM consortium author list.**
(PDF)

## Author Contributions

**Conceptualization:** Dorine M. Borensztajn, Irene Rivero Calle, Ian K. Maconochie, Ulrich von Both, Enitan D. Carrol, Juan Emmanuel Dewez, Marieke Emonts, Michiel van der Flier, Ronald de Groot, Jethro Herberg, Benno Kohlmaier, Emma Lim, Federico Martinon-Torres, Ruud G. Nijman, Marko Pokorn, Franc Strle, Maria Tsolia, Clementien Vermont, Shunmay Yeung, Dace Zavadska, Werner Zenz, Henriette A. Moll.

**Data curation:** Dorine M. Borensztajn, Nienke N. Hagedoorn.

**Formal analysis:** Dorine M. Borensztajn, Nienke N. Hagedoorn.

**Investigation:** Dorine M. Borensztajn, Nienke N. Hagedoorn.

**Methodology:** Dorine M. Borensztajn, Daan Nieboer.

**Supervision:** Henriette A. Moll.

**Writing – original draft:** Dorine M. Borensztajn, Werner Zenz.

**Writing – review & editing:** Dorine M. Borensztajn, Nienke N. Hagedoorn, Irene Rivero Calle, Ian K. Maconochie, Ulrich von Both, Enitan D. Carrol, Juan Emmanuel Dewez, Marieke Emonts, Michiel van der Flier, Ronald de Groot, Jethro Herberg, Benno Kohlmaier, Emma Lim, Federico Martinon-Torres, Daan Nieboer, Ruud G. Nijman, Marko Pokorn, Franc Strle, Maria Tsolia, Clementien Vermont, Shunmay Yeung, Dace Zavadska, Michael Levin, Henriette A. Moll.

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
