## [Decision Letter · Decision Letter 0]

26 Oct 2020

PONE-D-20-27296

Variation in hospital admission in febrile children attending the Emergency Department (ED) in Europe: PERFORM, a multicentre prospective observational study.

PLOS ONE

Dear Dr. Borensztajn,

Thank you for submitting your manuscript to PLOS ONE. After careful consideration, we feel that it has merit but does not fully meet PLOS ONE’s publication criteria as it currently stands. Therefore, we invite you to submit a revised version of the manuscript that addresses the points raised during the review process.

We look forward to receiving your revised manuscript.

Kind regards,

Tai-Heng Chen, M.D.

Academic Editor

PLOS ONE

Journal Requirements:

2.Thank you for your ethics statement:

'The study was approved by the ethics committees of the participating hospitals. The need for informed consent from individual patients was waived. In all the participating UK settings, an additional opt-out mechanism was in place. Patients were not directly involved in the design of this study.'

3.We note that you have indicated that data from this study are available upon request. PLOS only allows data to be available upon request if there are legal or ethical restrictions on sharing data publicly. For information on unacceptable data access restrictions, please see http://journals.plos.org/plosone/s/data-availability#loc-unacceptable-data-access-restrictions.

4. One of the noted authors is a group or consortium [PERFORM consortium]. In addition to naming the author group, please list the individual authors and affiliations within this group in the acknowledgments section of your manuscript. Please also indicate clearly a lead author for this group along with a contact email address.

Reviewers' comments:

Reviewer's Responses to Questions

**Comments to the Author**

1. Is the manuscript technically sound, and do the data support the conclusions?

Reviewer #1: Partly

Reviewer #2: Yes

2. Has the statistical analysis been performed appropriately and rigorously? 

Reviewer #1: No

Reviewer #2: I Don't Know

3. Have the authors made all data underlying the findings in their manuscript fully available?

Reviewer #1: Yes

Reviewer #2: Yes

4. Is the manuscript presented in an intelligible fashion and written in standard English?

Reviewer #1: Yes

Reviewer #2: Yes

5. Review Comments to the Author

Reviewer #1: 1. It is routine standard of care to admit all febrile children < 1 month, and in some institutions < 2 months. Including these patients may have biased the sample. I suggest the authors separate this group out for a subanalysis.

2. ED admission may also reflect the practice pattern (i.e., conservative vs liberal) of the admitting physician and culture at the hospital site. It also can reflect socioeconomic conditions surrounding each hospital experienced by the patient population. It can also reflect the resources (specialty care, ICU, ID) as stated in the methods description at each hospital. The hospital level factors controlled for are not sufficient to capture those important sources of variation. Therefore controlling for the 12 hospital clusters in the statistical analysis should be done - you say in your methods you 'analyzed variation between hospitals by age groups, ...' but was it only by those variables?

3. The conclusions are lacking - it's not sure what these results will add to the practice of emergency medicine, or to health services in general. If it is, as the authors say in the introduction, they wish to describe variation in admission patterns, then looking at variation between the 12 sites would be more illuminating.

4. Using PEWS is ok, although it is best known for how the change in PEWS value predicts critical illness, rather than one PEWS score snapshot in time. This should be mentioned in the limitations section.

5. Why were patients receiving IV antibiotics excluded? This would be a large amount of those admitted, and a reason for admission in itself.

6. Not all appropriate pediatric hospital admissions directly relate to disease severity - for example, febrile neonates who appear well and likely have a viral infection, or a cellulitis crossing a major joint - yet those are still appropriate admissions. Therefore the statements throughout the manuscript that 'overall admission rates reflect disease severity' are not appropriate.

7.

Reviewer #2: This is an interesting multicenter paper looking at variation in admission and management practices in febrile children across a network of pediatric EDs in Europe 2017-2018. They describe system differences, illness differences (final diagnosis), testing and other patient differences. The goal of this work is to reduce waste brought upon by unnecessary practice variation. It is timely in that the concern during seasonal flu (and this case COVID 19) has the possibility of really impacting health care resources around Europe and the world.

Some of my suggested edits may be to rewording due to clarity of language (pages refer to pdf #)

p 8 - Title - consider changing attending to evaluation in

p 12 - Abstract - The comparison of ED admission rates is complex in potentially being influenced by the characteristics of the region, ED, physician and patient. ….in order to use this to reduce health care utilization waste that is often due to practice variation.

p 14 - the MOFICHE study is a multicentre study evaluating or studying

- consider adding a sentence about admission as a proxy to severity may be less evidence based when the admission is very short or therapies not escalated.

- "routine" date - is not a term often used - is there a better one? routinely documented?

p 15 - is data collection manual or electronically downloaded?

p 16 - definitions are good, it is not clear if the providers used a case report form for their histories and findings to have standardized data OR if the data was extracted to fit what the provider filled out in the EHR. Please clarify.

p 17 - since you are partly testing diagnostic uncertainty's impact on admission - why not have the clinical diagnosis in the ED and then the discharge diagnosis in the cohort admitted to help understand what evolved?

p 18 - data quality - not clear if you are training the data extractors OR if you are working to improve the medical records data to prospectively obtain the data you are looking for.

p 19 - table 1 - was the question of medical care in the past 5 days asked of the patient or linked to actual visits? Referrals - - if a patient was told to come in by their provider and came by EMS, how is that coded? Vital signs - all patients were febrile or had a history of fever - - did you seek to try and get VS after treatment of the fever to look at change in VS?

p 21 - thoughts about putting a figure with the way you derive the standardized vs expected admission in the manusript? It feels like the concept is important to the work and having a visual regarding it would be good.

The rest of the paper is generally well written and it is fascinating regarding the UK 4 hour rule whihc may be contributing to the rata of admission and substantial proportion who were in for < 24 hours compared to the other centers. THe ICU admission variation does speak to the question of lack of standardization of ICU admission criteria in pediatrics more broadly, espcially since in centers in the US - both the utility of Medical Response Teams (and use of PEWS, for example) may lead to ICU admits that are short and without any or many interventions.

Just a comment on the provider variability - since there are PEM, pediatricians and both at some hospitals, it seems like a good question of how the amount of experience and the training / perspective of the groups may be important in what your next steps are to get "waste out of the system".

References - there are a good number of excellent ones

Tables and Figures - Overall these are quite good.

6. PLOS authors have the option to publish the peer review history of their article (what does this mean?). If published, this will include your full peer review and any attached files.

Reviewer #1: No

Reviewer #2: No

---

## [Author Response · Author response to Decision Letter 0]

8 Dec 2020

Dear Dr. Chen, dear reviewers

Thank you for your time, effort and expertise in reviewing our manuscript and taking it under consideration for publication.

We have adapted the manuscript according to the comments and we think that the manuscript improved after these important comments.

We have uploaded our response as a separate file.

---

## [Decision Letter · Decision Letter 1]

17 Dec 2020

Variation in hospital admission in febrile children evaluated at the Emergency Department (ED) in Europe: PERFORM, a multicentre prospective observational study.

PONE-D-20-27296R1

Dear Dr. Borensztajn,

We’re pleased to inform you that your manuscript has been judged scientifically suitable for publication and will be formally accepted for publication once it meets all outstanding technical requirements.

Kind regards,

Tai-Heng Chen, M.D.

Academic Editor

PLOS ONE

Reviewers' comments:

Reviewer's Responses to Questions

**Comments to the Author**

1. If the authors have adequately addressed your comments raised in a previous round of review and you feel that this manuscript is now acceptable for publication, you may indicate that here to bypass the “Comments to the Author” section, enter your conflict of interest statement in the “Confidential to Editor” section, and submit your "Accept" recommendation.

Reviewer #1: All comments have been addressed

Reviewer #2: (No Response)

2. Is the manuscript technically sound, and do the data support the conclusions?

Reviewer #1: Yes

Reviewer #2: Yes

3. Has the statistical analysis been performed appropriately and rigorously? 

Reviewer #1: Yes

Reviewer #2: Yes

4. Have the authors made all data underlying the findings in their manuscript fully available?

Reviewer #1: Yes

Reviewer #2: Yes

5. Is the manuscript presented in an intelligible fashion and written in standard English?

Reviewer #1: Yes

Reviewer #2: Yes

6. Review Comments to the Author

Reviewer #1: Thank you for revising the manuscript and incorporating reviewer feedback. The addition of age group breakdowns and improved description of the methods is very helpful for the reader.

Reviewer #2: The authors did a good job in both responding to the comments and feedback and in making edits. The only two areas I want to comment on are: writing style and next steps for this work. There is still some room for cutting and minor editing around the writing style to reduce the use of adverbs at the start of sentences that do not add to the work. The second is the question of what the specific next steps may be for the investigators to consider to reduce the variation of admission that do not use resources for additional management.

I would also call out that I was unaware of how the authors defined "normal" vital signs by age or other criteria would used to help distinguish severity or uncertainty.

Maybe some of the weaknesses of the study can just call out that retrospective studies do not have standardized collection of data in real time that helps discern what was present versus what was felt important to document. The question of the 4 hour rule is also an important question of whether it contributes to shorter ED lengths of stay but means some more unnecessary admissions.

Thank you for your turnaround of this work and attention to details on this revision.

7. PLOS authors have the option to publish the peer review history of their article (what does this mean?). If published, this will include your full peer review and any attached files.

Reviewer #1: No

Reviewer #2: No

---

## [Editor Report · Acceptance letter]

30 Dec 2020

PONE-D-20-27296R1 

­Variation in hospital admission in febrile children evaluated at the Emergency Department (ED) in Europe: PERFORM, a multicentre prospective observational study. 

Dear Dr. Borensztajn:

I'm pleased to inform you that your manuscript has been deemed suitable for publication in PLOS ONE. Congratulations! Your manuscript is now with our production department. 

Kind regards, 

on behalf of

Dr. Tai-Heng Chen 

Academic Editor

PLOS ONE